# Vulnerability of Coastal Areas Due to Infrastructure: The Case of Valencia Port (Spain)

**Vicent Esteban Chapapría *** and **José Serra Peris**

Ports & Coastal Engineering, Universitat Politècnica Valencia, 46022 Valencia, Spain; jserra@upv.es
* Correspondence: vesteban@upv.es

**Abstract:** The vulnerability of coastal areas is related to the existence and functionality of infrastructure. Ports have had increased activity in the last few decades due to growing needs of the market. At the same time, there have been huge changes in maritime traffic, and some ports are specialized in container traffic. The port in Valencia developed notably in the last expansions, in the 1980s and in the recent northern expansion. Valencia's port specializes in container traffic, and has become a Mediterranean leader and the metropolitan area is an important logistics center. Ports can create coastal erosion by altering wave patterns. The environmental effects of the port of Valencia were analyzed. The Spanish Mediterranean coastline as well as morpho-dynamic units were monitored. The solid transport capacity to the north and south of the Valencia port was estimated, and the effects of other infrastructure on sedimentary sources of beaches were also studied. The port of Valencia's barrier effect is responsible for the situation at the beaches to the north and south. This effect is total and impedes net sediment transport, predominantly to the south along the stretch of coastline. However, the port is not the only factor responsible for this situation, and the lack of continental sediments must also be considered. In addition, climate change has an influence on the behavior of the coastline. The vulnerability of the coast has increased due to changes in coastal morphology, variations in littoral transport rates, and coastal erosion. To promote sustainable port management, some correction measures, such as sand bypassing, dune rehabilitation, and dune vegetation, are proposed.

**Keywords:** coastal erosion; ports; sediments sources; reservoirs; sedimentary transport

## 1. Introduction

Changes in the characteristics of maritime traffic have created a highly competitive environment among shipping ports [1,2]. Between 1990 and 2007, before the last economic crisis prior to the pandemic, the average growth of the world's GDP was 2.3% and, in the same period, external commerce grew by 8.2% [3,4]. This was basically caused by the offshoring of industry in search of better productivity, entry into the market of new geographic areas, a reduction in transport costs due to technological advances and business organizations in the sector, the growing need of the markets for a supply of global products, and the increased acquisitive power of countries in economic transition, for example Eastern Europe, China, and the ASEAN countries [5–8].

The operators of container terminals, which form most of today's traffic, are concentrated in certain ports all around the world. The world's leading operator handles 12% of global business, and the top ten, together, account for almost 70% [9]. In just a few decades, these operators grew from local public centers to global and private centers. In addition to this phenomenon of concentration of terminals, all the important ones have become highly automated, although there are many uncertainties [10]. While ports have been obliged to follow suit, and as many of these ports are in coastal areas, they have a large environmental impact [3,11]. On the other hand, ports and terminals are also vulnerable to extreme weather events and sea-level rise due to climate change. Adaptation plans to climate change are being considered worldwide [12].

Coastal erosion is a sedimentary process that occurs whenever sedimentary transport away from the shoreline occurs and it is not balanced by new material being deposited onto the shoreline [13–17]. Many factors are involved in this event, in particular environmental and human factors. All over the world, coastal erosion is affecting the environment as well as economic activities. Additionally, global climate change and regional and local conditions are eroding the coasts of the world at troubling rates [18,19]. Some experiences of monitoring systems of coastal areas have been developed to improve the scientific knowledge and improve the tools for managing coastal hazards, thus reducing exposure to risks [14,20,21]. Coastal erosion in Europe is a major problem, where about 20,000 km of shoreline (circa 20% of the coastal territory) has already faced serious coastal erosion in the past, and about 15,100 km of shoreline are actively retreating or have been seriously affected by erosion [22,23].

Consequently, the objectives and goals of the developed research for the region are:

- To assess the state of coastal erosion;
- To determine the extent to which the coast around Valencia has increased its vulnerability due to the existence of the port and other infrastructure;
- To propose measures to correct problems, and the necessary adaptation to climate change;
- To promote adequate management of the port of Valencia.

## 2. Area and Methodology of Study

### 2.1. The Port of Valencia

The port of Valencia is in the central area of the Spanish Mediterranean coast (Figure 1). The history of this port is one of difficulties that had to be overcome to create the present installations, starting from a beach with shallow water and no protection against waves. The town center of old Valencia was built 3 km inland on the river Turia, which at that time emptied into the sea, and its waters were navigable up to the city walls.

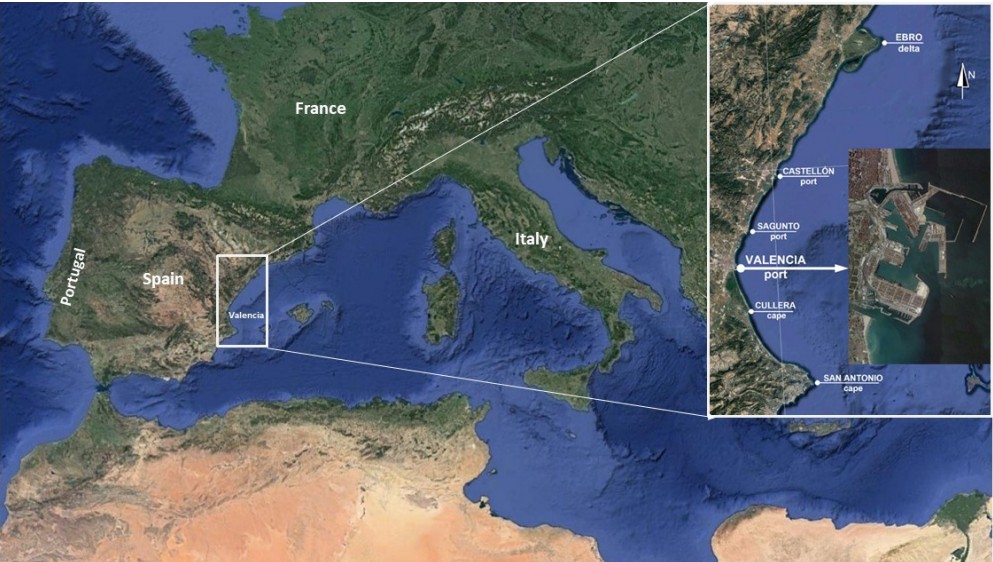

**Figure 1.** Valencia port location (prepared by the authors based on Google Earth).

The first harbor was constructed in 1483 and lasted until 1555. Activity in the port increased during the 14th and 15th centuries, and the need for a better harbor was felt as the sandy coast could not offer much protection to boats; the currents and storms eliminated the succession of piers that were built.

In 1686, the city council began to build the first stone pier (Figure 2), but this was never finished and was eventually destroyed. Other projects were subsequently suggested but were never built. In 1792, the engineer and naval captain Manuel Miralles started work on a new port (Figure 3), but the work was interrupted by the War of Independence in 1808.

In 1826, improvements were made to give it its present configuration. The East and West Quays were built during the 19th century, and they formed a pentagonal harbor that gave rise to the plan of the present port of Valencia.

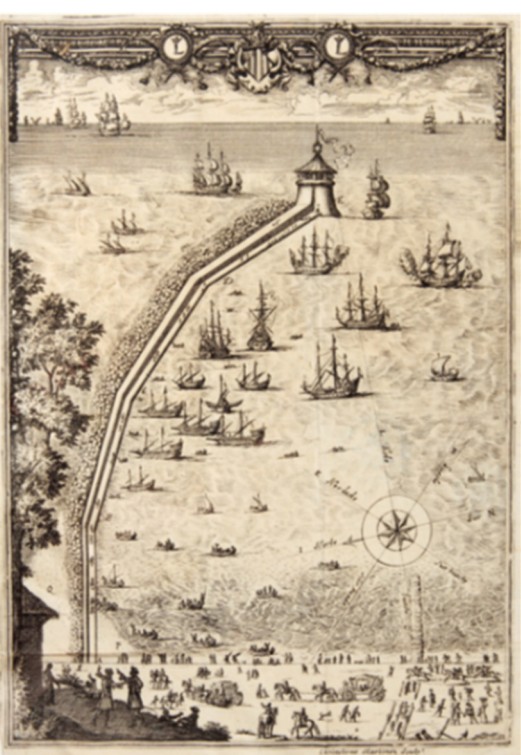

**Figure 2.** Güelda Valencia port plan, 1686 (Source: Valenciaport.com accessed on 10 June 2021).

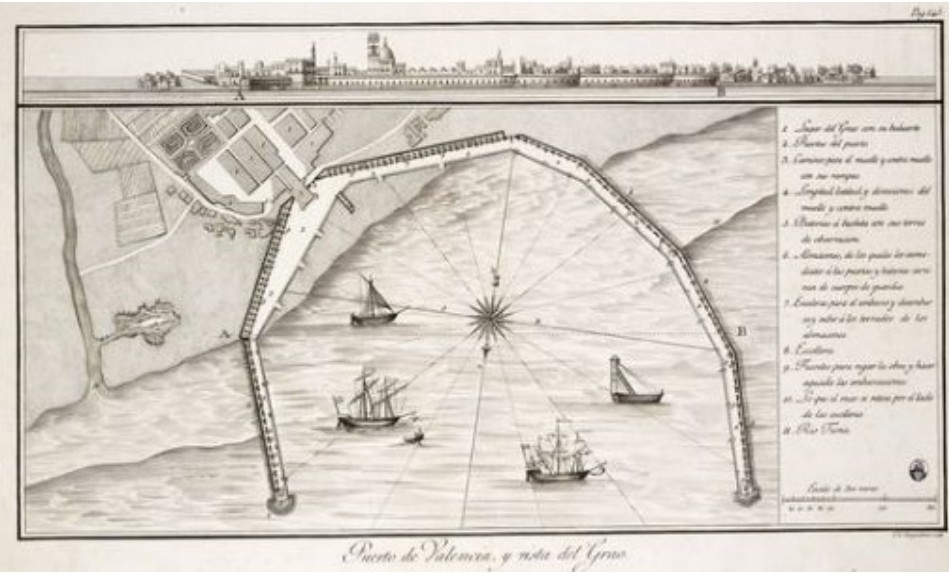

**Figure 3.** Manuel Miralles' Valencia port plan, 1792 (Source: Valenciaport.com accessed on 10 June 2021.

Figure 4 shows the general plan and the layout of the port in 1930, 1950, 1974, and 1994, the years in which the port could first be considered to offer some protection to moored ships, and the progress of the breakwater towards the open sea and to the south to make way for the new path of the Turia river to the sea.

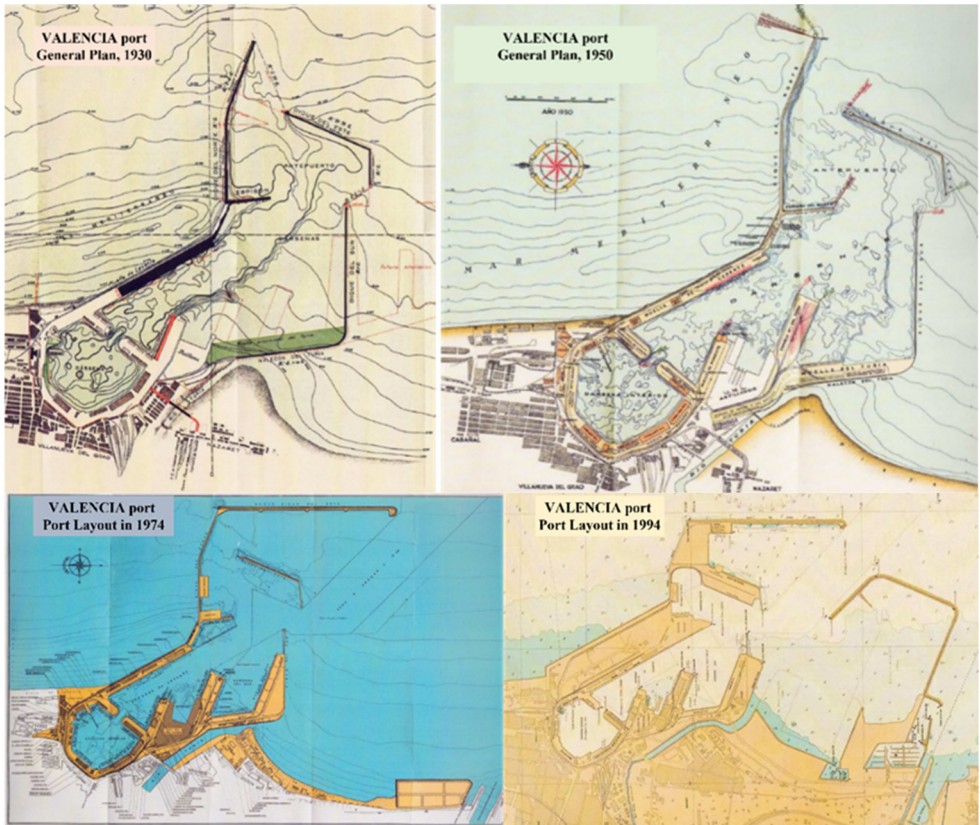

**Figure 4.** Valencia port in 1930, 1950, 1974, and 1994 (Source: Valenciaport.com accessed on 10 June 2021).

Figure 5 shows the progress of the interior works, the expansion of the port, and the so-called northern expansion.

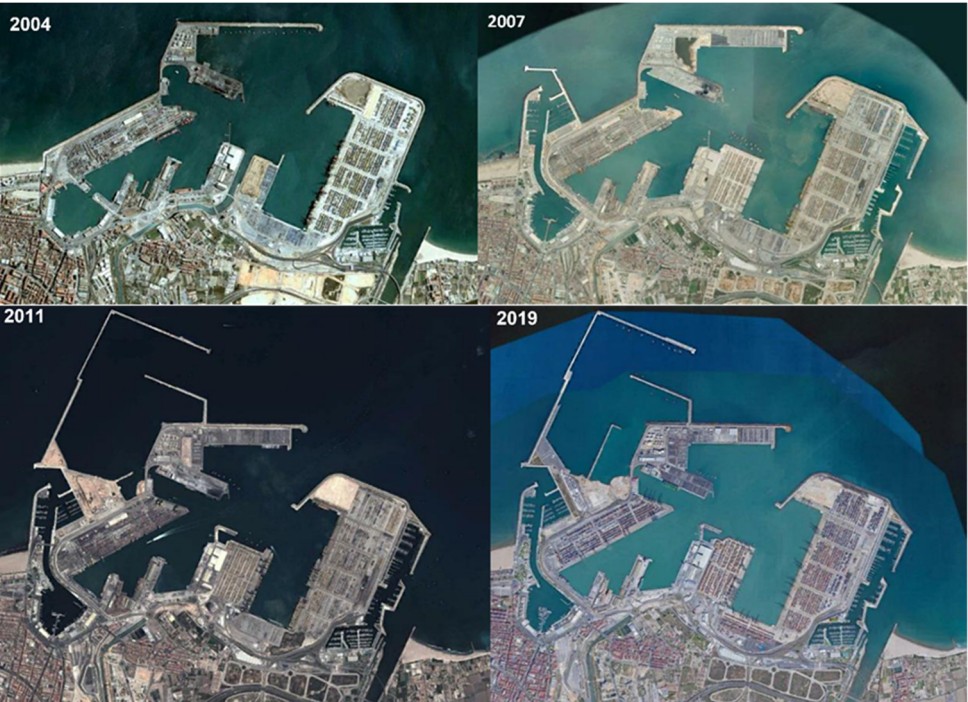

**Figure 5.** Valencia port layout in 2004, 2007, 2011, and 2019 (Source: Valenciaport.com accessed on 10 June 2021).

### 2.2. The Port and the Transformation of Valencia and Its Metropolitan Area

A great deal of global maritime traffic now passes through the Mediterranean, and Valencia is situated in the middle of the Spanish Mediterranean coastline. Valencia and the surrounding region have experienced enormous growth in recent decades. Fifty years ago, the city had little more than half a million inhabitants and did not extend much further past the old city walls than it had had in the 19th century. The Turia was diverted from its original course through the city center in 1969 to prevent occasional catastrophic floods. In 1957 a flash flood devastated the center and carried off many buildings, including historic ones, so there was a housing shortage and transportation was difficult; it also lost some of its parks. In the last 50 years, the population has grown to one and a half million, and Valencia now belongs to a group of 30 European second-scale cities, just below the first group, or the so-called decision-making group. The port has been transformed by profound changes to its installations, and today it is the country's leading container, general cargo, and ferry terminal.

Transformation of the urban world, the networking of cities, and the multi-pole system with its great concentration of world leaders outside Europe have generated large changes: " . . . the infrastructure creates cities just as the city contributes to making cities. There is a well-known feedback between transport infrastructure improvements and the mobility generated from social relations and economic activities: infrastructure creates cities as well as cities create infrastructure" [24].

The growth of the port of Valencia continued until it became one of the Mediterranean leaders, together with the metropolitan and national transport networks, and it has become an important logistics center. The zone has great attraction for generating production facilities. In 1970, the Ford company decided to build a car plant there and its subsequent growth and consolidation had a lot to do with the infrastructural capital that the zone now has to offer, where the port of Valencia plays a leading role. One of the area's highlights is its remarkable growth and development, including the port of Sagunto, which together form a docking system. Each has been developed to exploit the capacities of the port of Valencia as a mixed hub to optimize costs of scale and the mixed volume of local imports and exports, with Valencia as the leader. In 2018, it handled 92.4% of all goods traffic, 98.9% of containers, 95% of passengers, and 65% of new vehicles. Forty percent of containers containing Spanish exports and 60% of the containers with exports from Madrid are handled by Valencia's port [25].

### 2.3. Evolution of Freight Traffic in the Port of Valencia

Valencia's port has become one of the leading international ports and has been specifically planned to specialize in containers. The first of these arrived in 1970 and the first privately financed crane designed to handle them was installed in 1972. In 1980, the port handled 8 million tons of cargo and 118,000 containers. Thirteen years later, these figures had grown to 10.4 m tons and 385,000 containers.

The growth increased dramatically over the subsequent 25 years, and the installations were expanded to cope with it. Between 1993 and 2018, container traffic was 13 times larger and the cargo tonnage increased by a factor of seven. The evolution experienced in these years is shown in Tables 1 and 2. Compared to other Mediterranean ports, this evolution was considerable and put Valencia in the first position in 2018 (Table 2).

In 2019, Valencia was the fifth European port in container traffic and the first Spanish port in terms of (a) total container traffic; (b) import/export containers; (c) it was second in total traffic, and (d) it was first in terms of general cargo, roll-on roll-off, and automobiles. It is a leading international ocean port, and Valencia has a well-developed economic infrastructure. The total added value of the port of Valencia in 2016 was an economic impact of more than EUR 2499 M, or 2.09% of the total of Valencia's community. Valencia's port has 38,866 employees, or 2.09% of all jobs in Valencia's community, who receive EUR 1250 M in salaries, or 2.62% of all salaries in the community. The gross earnings of all the

companies directly employed by the port are EUR 1076 M, or 2.34% of the community's total earnings [23].

**Table 1.** Evolution of cargo handled by Valencia's port (1980–2018) (Source: compiled by the authors from APV records [25]).

| Year | M t * | M TEUs ** |
|---|---|---|
| 1980 | 8.0 | 0.12 |
| 1993 | 10.4 | 0.39 |
| 1998 | 20.5 | 1.05 |
| 2004 | 37.9 | 2.15 |
| 2007 | 53.6 | 3.04 |
| 2013 | 65.0 | 4.33 |
| 2018 | 76.6 | 5.18 |

* M t: millions of tons; ** M TEUs: millions of Twenty-foot Equivalent Unit containers.

**Table 2.** Evolution of container traffic in different Mediterranean ports (source: compiled by the authors from UNCTAD Report [26]).

| 1990 | | 2000 | | 2018 | |
|---|---|---|---|---|---|
| **Port** | **M TEUs** | **Port** | **M TEUs** | **Port** | **M TEUs** |
| Marseilles | 481 | Gioia Tauro | 2652 | **Valencia** | **4832** |
| Algeciras | 474 | Algeciras | 2009 | Algeciras | 4381 |
| La Spezia | 454 | Genoa | 1500 | Piraeus | 4060 |
| Barcelona | 447 | Barcelona | 1364 | Marsaxlok | 3150 |
| Livorno | 416 | **Valencia** | **1308** | Barcelona | 2969 |
| **Valencia** | **387** | La Spezia | 905 | Genoa | 2638 |
| Genoa | 310 | Marseilles | 726 | Gioia Tauro | 2449 |
| Gioia Tauro | 0 | Livorno | 478 | Mersin | 2328 |

Three container terminals were operating in 2019 in the port of Valencia: NOATUM-COSCO, MSCT/TIL, and TCV/APMTV, with an estimated capacity in 2017 of 6.1 M TEUs, which are, at present, working at 81% capacity. The port's turnover is estimated to double in the next 30 years, from 5.2 TEUs in 2018 to 10.5 M in 2050, and a new terminal is considered necessary in the northern sector, where breakwaters were built between 2008 and 2012. This new terminal will be a joint public–private project. It should be remembered that the effects of port infrastructure have important consequences for coastlines [25].

*2.4. Methodology of Study*

The methodology used aims to establish the causes of coastal erosion. To achieve coastal erosion management [27], coastal erosion research, especially on vulnerable areas, could be based on risk assessment [27–29]. To characterize coastal conditions around the port of Valencia, the following methodology was used [30,31]:

- Sediments sampling at dry and submerged beaches. Granulometric tests of the sampled sediments were developed once a sample was washed and organic matter was removed using ASTM sieves;
- The determination of the carbonates of the sampled sediments;
- The mineralogic analysis of fractions of ASTM series and analysis of the existence of blue quartz in fractions;
- Sediment characterization (including granulometric, carbonate portion, and mineralogic tests), allowing the establishment of potential sediment sources;
- The sedimentary transport by waves and storm surges was estimated. The wave climate in the area was established according to [32] from the following data sources:

- The Valencia tide gauge (REDMAR network) managed by *Organismo Público Puertos del Estado* (OPPE), an hourly data series provided by the Maritime Climate Program and including data since 1992.
- The GOS 2.1 (Global Ocean Surges) reanalysis database [32] to determine the storm surge. The numerical model used in the GOS reanalysis is the ROMS (Regional Ocean Modeling System) three-dimensional model developed by Rutgers' Ocean Modeling Group (http://marine.rutgers.edu/po/index.php?model=roms accessed on 16 September 2020).
- Relationships between the different sea reference levels were taken into account.
- Wind conditions (mean and extreme annual conditions) were determined, obtained through dynamic downscaling using the WRF-ARW 3.1.1 model (weather research and forecasting and advanced research dynamical solver) from the ERA-Interim atmospheric reanalysis [33] developed by the European Center for Medium-Range Weather Forecasts (ECMWF).
- Wave data at deep water conditions, also from the GOW 2.1 (Global Ocean Waves 2.1 [32]. This reanalysis includes data since 1989, has an hourly temporal resolution, and has a spatial resolution of 0.125° along the Mediterranean. The numerical model used for the generation of the reanalysis was the Wave Watch III model developed by NOAA/NCEP. The SWAN-OLUCA mixed numerical model for the wave propagation to the coastline was applied [32] to obtain the breaking currents.

● Aerial and satellite views were also analyzed [26].

The longshore transport rate was estimated [34–36] using the following equation:

$$Q \,(\mathrm{m^3/year}) = 2027 * 10^6 * H_0^{5/2} * \mathrm{sen}(2\alpha_0) * \cos(\alpha_0)^{1/4} * K_p * K_g$$

The used methodology seeks to obtain results from the combined analysis of all the information. Integrated analysis offers advantages in obtaining comparisons and simulations of dynamic situations in order to explore possible strategies to overcome erosion and, thus sustain economic growth, minimize population risk, and maintain biodiversity [15].

## 3. Results

### 3.1. The Spanish Mediterranean Coast

The Spanish Mediterranean coastline is generally in the direction of S60° E, is fairly straight and low-lying without natural indentations, and has a considerable amount of port infrastructure coastal protection systems, including jetties, dykes, and breakwaters. It suffers considerable erosion due to the north–south movement of materials and the absence of sediments from the north, mainly due to the presence of docking installations and other construction. The beaches are mostly composed of $D_{50} = 0.25$ mm sand and occasionally gravel close to river mouths. Many bathymetric studies have detected submarine materials, mainly composed of fine sand of $D_{50} = 0.15$ mm [32].

The port of Valencia is situated in the center of a natural morpho-dynamic unit known as the Valencia Oval, confined on the north by the Ebro Delta and Cape St. Anthony to the south, both are natural and total barriers to sedimentary transport. The complete front is a continuous coastal formation (Figure 1). Due to its coastal characteristics, and its lack of natural shelter, the ports had to be won from the sea, and some form total natural barriers due to their considerable size, as in the case of Castellón. The natural Valencia Oval (Figure 1) is thus composed of three artificial morpho-dynamic units: Castellón (between the Ebro Delta and the port of Castellón), Valencia (between those of Castellón and Valencia), and Cullera (between the Port of Valencia and Cape St. Anthony).

There are also partial, normally small, barriers, as well as other infrastructure, on the coastline, including the ports of Vinaroz, Benicarló, Peñíscola, and Oropesa. In the Valencian artificial morpho-dynamic unit, the partial barriers are formed by the ports of

Burriana, Siles, Sagunto, Puebla Farnals, and Saplaya. In the Cullera artificial morphodynamic unit, they are the Perelló river mouth, the Júcar river in Cullera, the ports of Gandía, La Goleta, and Denia, and various coastal defense systems. The port of Valencia divides both units, so that the beaches to the north belong to Valencia and those to the south belong to Cullera. The Valencia Oval can be considered as a single coastal system that receives sediments from the Ebro river.

### 3.2. Analysis of Sediment Transport in the Valencia Oval

Some advantages have been determined from a combined comparison of results [29]. The net solid transport capacity to the north of Valencia has been estimated to be between 100,000 and 130,000 m$^3$/year in a southerly direction. The same study analyzed the accumulated sediments over diverse periods. Between 1947 and 1957, the shoreline advances indicated that sediment transport on the beaches to the north of the port of Valencia had values of between 40,000 and 50,000 m$^3$/year. Between 1957 and 1965, the shoreline advances indicated sediment transports of 130,000–160,000 m$^3$/year, and between 1965 and 1972 they were 100,000–130,000 m$^3$/year, these values were the same between 1972–1977. The mean annual accumulation of sediments between 1981 and 2001 are estimated to be between 27,000 and 30,000 m$^3$/year.

Research has been carried out [37,38] based on aerial photos, and determined the volumes of additions and losses of sediments on the coastline. Figure 6 shows the evolution of the coastline, calculated by photogrammetry, between the port of Saplaya and the port of Valencia. This shows the coastline to have expanded considerably between 1957 and 1965. In one 5 km stretch, a volume of 4,470,219 m$^3$ eroded between 1947 and 2000, and there was an accumulation of 1,082,632 m$^3$, so that it can be concluded that the transport capacity was approximately 64,000 m$^3$/year.

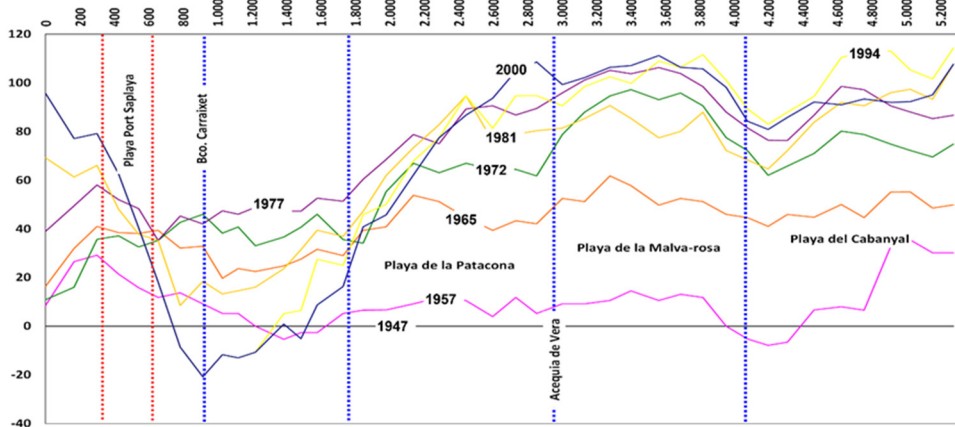

**Figure 6.** Shoreline changes rates and coastal evolution to the north of Valencia's port (prepared by the authors).

Figure 7 shows the situation to the south of the port, between Valencia and the Gola del Pujol. In this case, the almost 8 km of coastline are in recession and sediment transport is estimated to be 210,000 m$^3$/year. Considering both shorelines to the north and south of Valencia, a weighted average of 150,000 m$^3$/year can be established in a southerly direction, which is consistent with the previous estimation, so the net solid sediment transport capacity can finally be estimated at between 100,000 and 150,000 m$^3$/year.

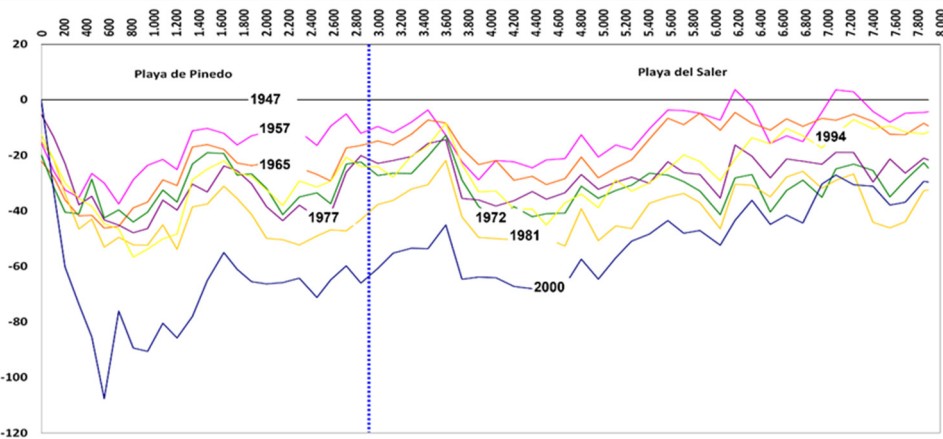

**Figure 7.** Shoreline changes rates and coastal evolution to the south of Valencia's port (prepared by the authors).

### 3.3. Sedimentary Sources

Rivers have historically been the sedimentary sources of materials for Valencia's beaches. Only the Mijares river remains as a primary source of materials. There are secondary providers, the Palancia river and the Barranco del Carraixet. Figure 8 shows the most important sources of sediments (sand and gravel) for the formation of beaches in the Oval as well as the materials formed by the erosion of the low cliffs.

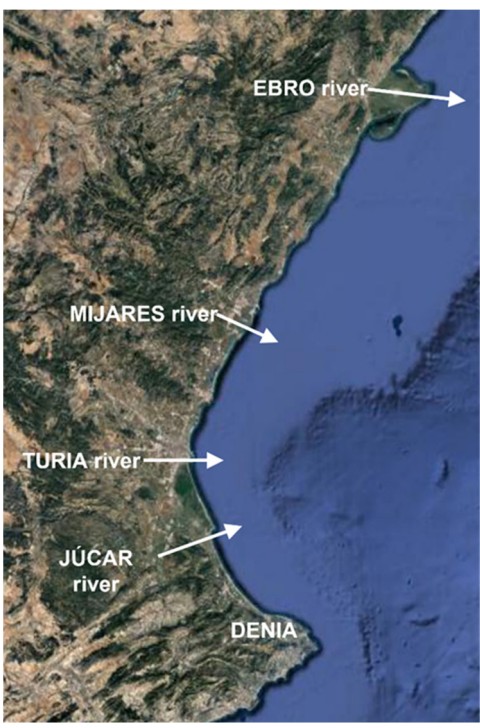

**Figure 8.** Main sedimentary sources (prepared by the authors based on Google Earth).

Table 3 gives the rivers for which figures are available, including the names, gauging stations, locations, river flows, and distance from the station to the river mouth.

**Table 3.** River data from different agencies (prepared by the authors).

| River | Gauging Station | Location | Maximum Flow (m3/s) | Minimum Flow (m3/s) | Distance from Coast (kms) |
|---|---|---|---|---|---|
| *Ebro river Agency* | | | | | |
| EBRO | Tortosa | Tortosa | 152.00 | 103.00 | 46,100 |
| *Jucar river Agency* | | | | | |
| Castellón Morpho-dynamic Unit | | | | | |
| SÉNIA | Ulldecona Dam | La Pobla de Benifassà | 1.77 | 1.30 | 36.76 |
| CERVERA | Cervera del Maestre | Cervera del Maestre | 0.00 | 0.00 | - |
| *Valencia Morpho-dynamic Unit* | | | | | |
| MIJARES | Azud Sta. Quiteria | Villareal | 5.88 | 4.27 | 9.35 |
| PALANCIA | Embalse Regajo | Jérica | 1.70 | 1.37 | 48.00 |
| CARRAIXET | Carraixet ravine | Bétera | 0.00 | 0.00 | - |
| *Cullera Morpho-dynamic Unit* | | | | | |
| TURIA | La Presa | Manises | 0.00 | 0.00 | - |
| JÚCAR | Azud de la Marquesa | Cullera | 9.27 | 4.40 | 4.23 |
| SERPIS | Vernissa ravine | Rótova | 0.85 | 0.85 | - |
| GALLINERA | Adsubia | Adsubia | 0.01 | 0.01 | - |

## 4. Scientific Discussion

The vulnerability of the coast due to climate change is manifested in: (a) increases in flood levels; (b) beach profile changes, erosion, and limit depth of affectation increases, and (c) changes in coastal morphology, variation in littoral transport rates, and coastal erosion. Ports can create coastal erosion by altering wave patterns. Studies attempts to clarify environmental effects in managing port-induced coastal erosion occurring at beaches that are intensively used by the population.

### 4.1. Sediments and Their Sources

The lowest parts of the coast or sandy beaches formed by the accumulation of sediments carried by currents can be found along the entire coastline, but the question is "what is the origin of these sediments?" There are a large number of sources, rivers being the biggest contributors. On the Mediterranean Coast, more than 90% of the sand and gravel comes from rivers, torrents, and streams, while the rest comes from erosion materials from cliffs, in addition to vegetable matter. Figure 8 shows the main sources of sediments in the Valencia Oval. In most cases rivers carry little solid matter and some are overexploited.

Apart from the Ebro, with a mean flow of more than 100 m$^3$/s, the other rivers for which figures are available have mean flows of less than 2 m$^3$/s (1.76 m$^3$/s). In some cases, they are protected to conserve their flora and fauna. No river exists that can guarantee a supply of sediments to sustain the sandbanks, which depend on continental supply. There are quite a few smaller rivers for which no figures are available and whose mouths are usually blocked by sandbars and only have flows after heavy rains, as in the case of the wetlands, and some have locks to regulate their levels.

It can thus be seen that, according to the river flows and conditions, there is no large sediment transport capacity. There are also other problems that affect the transport of sediments from rivers, such as the overexploitation of aquifers, the maintenance of riverbanks, and the presence of dams. The construction of a dam to store water means it will also store sediments that will never reach the river mouth and thus cannot provide them to the coast. Figure 9 shows dams with their gauging stations to the north of Valencia's port in the Valencia morpho-dynamic unit.

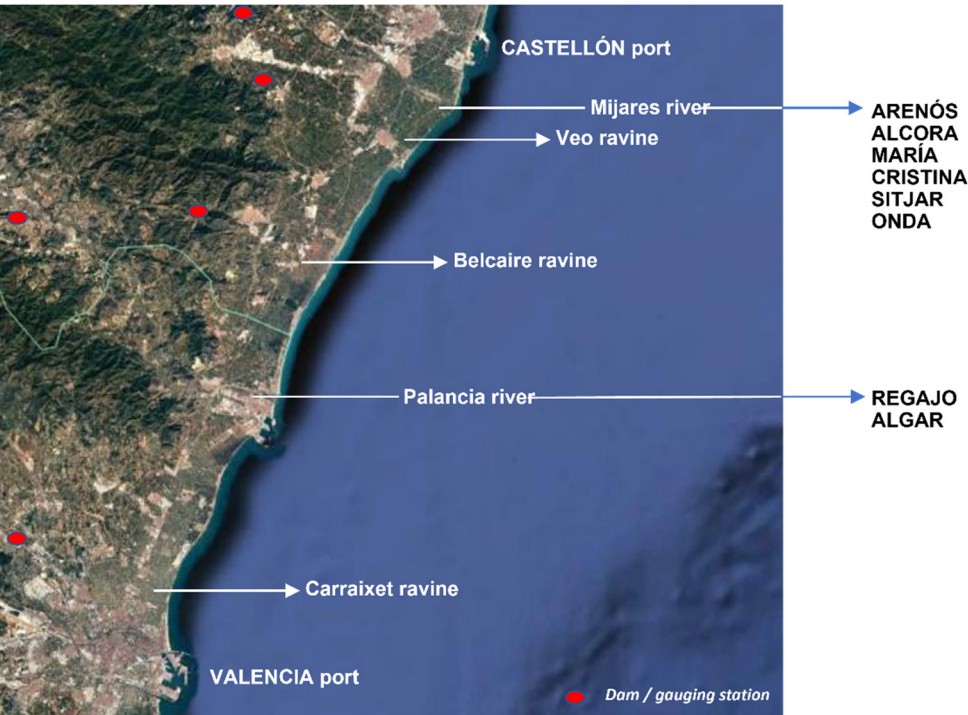

**Figure 9.** Sedimentary sources and reservoirs' locations in the Valencia morpho-dynamic unit (prepared by the authors based on Google Earth).

Results of the mineralogic analysis of the sediments in the Valencia Oval show that the most important contributing rivers are the Ebro, Mijares, Turia, and Júcar. Sediments from the Ebro reach the most southerly beaches in the Oval. The Mijares previously fed beaches to the north and presently the Turia and Júcar provide almost nothing, based on the results of the analysis of tourmaline, which is only present in the Ebro and Míjares. Some studies evaluated the volumes of sediments in dams and hydrographic basins [5], giving details of the landslides in the Ebro Basin, along with the names of the dams, their initial capacities in $hm^3$, the volume of the landslides in $hm^3$, the year of the evaluation, and the dams' percentage of lost volume. The average landslide volume was estimated to be 0.54 $hm^3$/year.

The same data for the Júcar Basin are given [5], and, in this case, the average landslides were estimated at 147,049 $m^3$/year, much less than for the Ebro, due to its much smaller rivers. Finally, an estimate of the landslides in all Spanish basins in 2025 are also given. The estimated value for the Ebro is 1330.75 $hm^3$ and it is 293.69 $hm^3$ for the Júcar.

The estimated landslide figures for the Valencia morpho-dynamic unit are 52,614 $m^3$/year and for that of Cullera, 224,308 $m^3$/year. The rivers in the latter unit are to the south of Cape Cullera, from which it can be concluded that their contributions do not reach the beaches just south of the port of Valencia. In the case of the Valencia Unit, the possible contributions come from north of the port of Sagunto. There are two small harbors between there and Valencia (Puebla Farnals and Port Saplaya) and a stretch of the Playa Norte coastline with a breakwater, so that reduced contributions of the rivers with mouths in this unit have little effect on the beaches to the north of Valencia. In regard to sources of sediments, it can generally be said that almost none of these are from continental sources that guarantee a sustainable supply to beaches around the port of Valencia.

### 4.2. The Beaches to the North and South of the Port of Valencia

The beaches to the north of Valencia have different types of patterns due to the barrier effect of the port on the coastline, in which the sediment is transported towards the south. The beaches of Malva-rosa and Cabanyal in the north (Figure 10) form a coastal front with no continental contribution and receive sediments from the north–south current. Malva-

rosa is eroding, while Cabanyal clearly receives sediments, according to studies carried out between 2008 and 2015.

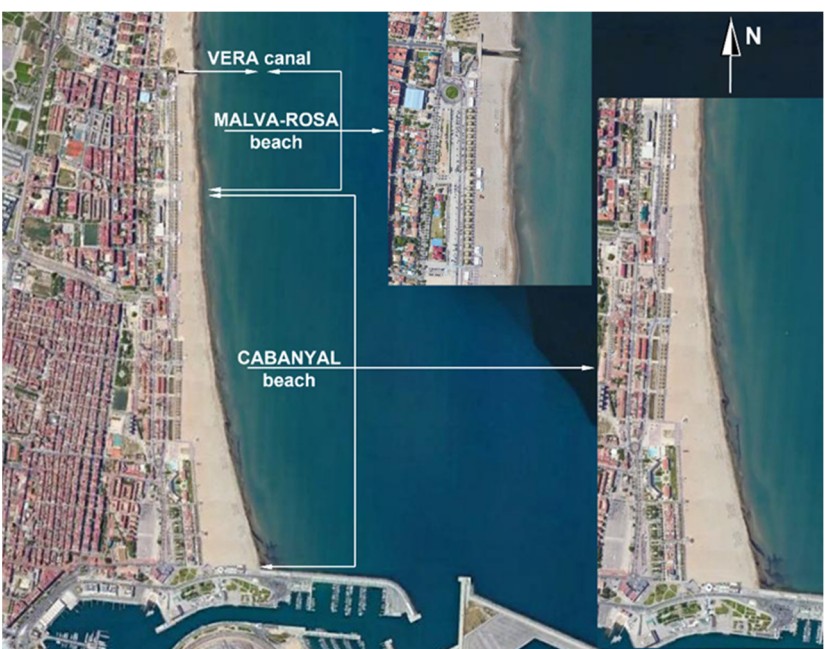

**Figure 10.** Beaches to the north of Valencia's port (prepared by the authors based on Google Earth).

To the south (Figure 11) there are a large number of stretches that behave differently. Pinedo Beach, just south of the port, and within its shelter, receives sediments, while the rest of the beaches in the stretch are being eroded, with some being apparently stable but with occasional periods of erosion.

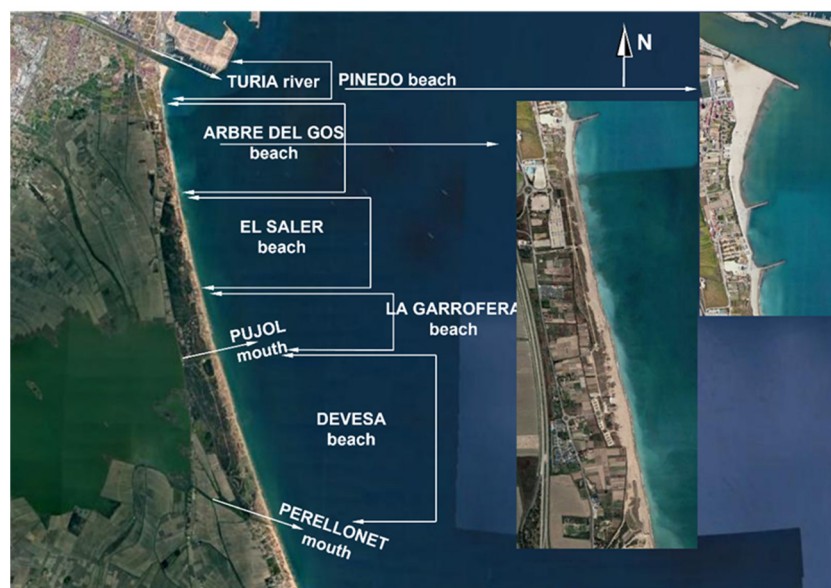

**Figure 11.** Beaches to the south of Valencia's port (prepared by the authors based on Google Earth).

*4.3. Effects of the Port on the Coast*

It can thus be seen that the port of Valencia's barrier effect has, for several decades, been responsible for the situation of the beaches to the north and south. This effect is total and impedes net sediment transport, predominantly to the south, along this stretch of coastline. The sands are thus carried further south and are deposited on beaches

downstream, while those upstream are in recession; however, the port is not the only factor responsible for this situation, as the lack of continental sediment must also be considered. It should be remembered that the largest contribution to the coastline comes from rivers and, in this zone, many water resources are overexploited to supply farmers and industry, and in the 1970s a housing estate was built just to the south of the port, which prevented the transport of a considerable volume of sand. Additionally, climate change will have an influence on the behavior of the coastline. The conclusion can thus be drawn that the port of Valencia is partly responsible for the recession of beaches to the south, but not entirely. Public opinion considers that the port is mainly responsible, and that the solution to recover southern beaches is to eliminate it; curiously enough, though, nobody blames it for the accretion of sediments on those to the north.

*4.4. Corrective Measures for Coastal Protection*

Coastal erosion management is defined as an interactive, dynamic, and multidisciplinary approach to cope with the coastal erosion processes [27]. Valencia's port has become one of the Mediterranean leaders and the Valencia metropolitan area is an important logistics center; both have to be protected. The most common practical strategies for managing coastal erosion are [23,39]:

- Protection of populations, economic centers, and vulnerable natural areas by using hard structures and/or soft protection measures;
- Accommodation through occupying sensitive regions, but acceptance of a higher degree of flooding, erosion, or other hazards by changing land use, construction methods, and improving preparedness;
- Planned retreat by removing structures in developed areas, resettling inhabitants, and requiring new development to be set back from the shore;
- Use of ecosystems influencing processes related to coastal erosion by means of the creation and restoration of coastal ecosystems, such as wetlands, biogenic reef structures, seagrass beds, and dune vegetation;
- Doing nothing and allowing property loss when protection is not possible, or if the accommodation and retreat option does not exist.

Most conventional coastal protection or coastal erosion mitigation methods are difficult and expensive to implement [17]. Additionally, they are not eco-friendly. Most of the difficult options have been found to interfere with littoral dynamics and sedimentary processes and cause downdrift erosion. The current trend in coastal erosion mitigation is to opt for soft, innovative, and proactive methods. Soft options are mostly eco-friendly and are currently being used in place of hard options all over the world. Soft options to be considered in Valencia area are sand bypassing, dune rehabilitation, and dune vegetation. They can correct the effects caused by Valencia's port on beaches. In addition, they can also prevent sand loss caused by stronger storms due to climate change.

## 5. Conclusions

The port of Valencia clearly forms a total barrier to sediment transport and its effects are defined. It has a distinct geometric shape that gave rise to the formation of the beaches of Cabanyal and Malva-rosa. This situation should mean that these beaches are in accretion, but this would require contribution by sediments from the north. The lack of a natural supply means that the sediments that reach them do not come from outside or distant regions. This is due to the existence of barriers to the north, e.g., the Ports of Castellón, Burriana, Siles, Puebla Farnals, Port Saplaya, breakwaters, marinas, the defense systems between Castellón and Valencia, and the absence of sediments from rivers.

The peculiar shape of Valencia's port formed the beach of Pinedo in its shelter to the south, while the rest of the coastline is in recession. Sand is accreting to its immediate south, but outside the port's sheltered zone there is a tendency to recede. The final conclusion that can be drawn is that the coast, due to the breakwaters to the north of the mouth of the Turia, produced two fronts with no problems of recession; however, the lack of sediments

from the rivers along this stretch, together with other actions, have led to a local front to the north in accretion due to the port, while preventing accretion to the south due to the loss of the sediment transport from the rivers. Together with the barrier effect, this has produced a front in general recession. Consequently, coastal vulnerability has increased due to the port and other infrastructure.

Modern ports play a key economic role and require the adequate management of infrastructure, road networks, and logistics. To correct the effects from the port of Valencia on beaches, it is necessary to promote soft and innovative solutions: sand bypassing, dune rehabilitation, and dune vegetation. These solutions will also work to prevent coastal erosion due to stronger storms caused by climate change.

**Author Contributions:** Both authors (V.E.C. and J.S.P.) all statements. All authors have read and agreed to the published version of the manuscript.

**Funding:** This research received no external funding.

**Conflicts of Interest:** The authors declare no conflict of interest.

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
