# Peer review of "Vulnerability of Coastal Areas Due to Infrastructure: The Case of Valencia Port (Spain)"

_land, doi:10.3390/land10121344_

Round 1

Reviewer 1 Report

The manuscript presents an interesting topic: "Vulnerability of coastal areas due to infrastructure". Unfortunately, as is, the manuscript resembles more a news magazine article or history lesson, rather than technical or scientific article. The history lesson on the evolution of the port of Valencia is really appreciated. The core of the manuscript should be section 2.4 "Methodology of the study", yet there is barely any description of a methodology here. A list of tests and parameters doesn't constitute a methodology.

How does each test fits into the study?

How was sedimentary transport estimated?

What were the data sources of waves and storm surge?

Without clearly stating what the actual methodology was, the "Results" section takes us again through yet another history lesson of freight traffic in the port of Valencia... and of the "Spanish Mediterranean coast"? How these constitutes results?

The authors proceed to discuss results from a study by LPC conducted back in 2004. Who are LPC? Where are the references?

This manuscript requires a complete rewrite. Authors need to demonstrate relevant technical contributions, including scientifically sound methodology, result, and discussion.

Author Response

Please, see attachment.

Yours sincerely

Reviewer 2 Report

The work can be published with some small contributions.
This study talks about the influence of the port of Valencia on the configuration of the coast, causing a modification of the erosive phenomena.
The authors only mention the influence of climate change on the coastline, nor do they report that corrective measures could be applied to the changes caused, for example the dune vegetation, which can prevent the loss of sand due to the strong storms that exist as a result of the climate change.
The authors are advised to propose corrective measures that protect coastal areas.

The work is correctly written, in which it is expressed as the infrastructure of the port of Valencia, Spain influences the coastline, introduction, methodology, results, discussion and conclusions are apparently correct; The diagnosis made on the problem is good, however I have not seen clear proposals on how to mitigate the action of this infrastructure on the coastline, for this reason I advise the authors to include a section with proposals on how the action of this could be avoided infrastructure on the environment.

Author Response

Please, see attatchment.

Sincerely yours

Reviewer 3 Report

In this article the authors analyzed the vulnerability of the Port of Valencia owing to the existence of infrastructure, by means of monitoring the coastal morphodynamics and sedimentary dynamics.

The study is interesting and well done. The information generated is useful to promote adequate management of the port of Valencia. However, it was not clear to me if vulnerability was directly addressed. Indeed, the study involved a great deal of effort and would be adequate for Land.

My only concern is that the article does not focus on its main goals. From the title and introduction, one would expect to read a study on the vulnerability of the coast of the Port of Valencia in response to coastal infrastructure. But this is not exactly the case. Vulnerability is not addressed directly. And, besides coastal infrastructure, the authors discuss the sediment deficit problems associated to dams. But there are no results or data on dams. The discussion and conclusions are not really such, but more results. This needs to be worked thoroughly and stay focused on the goals of the study, while briefly mentioning the caveats and strengths of the study performed.

In brief, I think that this is an interesting paper that deserves to be published because it contributes to assessing coastal vulnerability owing to built infrastructure. However, it needs to be condensed. The discussion needs to stay more focused and the conclusions need to be rewritten. Thus, I think that the ms. could be accepted for publication after major revision.

Specific comments follow:

Abstract

The abstract needs some editing. For instance, it is not clear the predominant activities in the Port of Valencia. Also, what environmental effects were analyzed? And the results are not summarized in the abstract. The “take-home message" should be clearer and stronger.

Introduction

The introduction is very interesting and nicely written. However, the goals and objectives need to be more clearly stated.

Area and methodology of study

Figure 1 should show the location of Spain and the location of Valencia in Spain, so that any reader unfamiliar with Spanish geography clearly understands the location of the study site.

The historic description of the Port of Valencia and corresponding images are very interesting.

Methods

For clarity, it would be better if Figure 1 showed the location of the study site, in a smaller scale, showing the USA-Canada border. This is useful for those potential readers not familiar with the USA geography. Also, please explain what the polygons represent.

Line 133. How did the authors study environmental economics of Valencia? What were the variables analyzed? This is not clear.

Lines 133-137 are difficult to follow. Please rewrite.

Results

The tables need their legends at the top of the tables, not at the bottom.

Table 1. Please explain what M t and M TEU mean. Are these yearly values?

Table 2. Why do you have TEUs in 1990 and 200 and Conts in 2018? You should use the same units. They need to be explained in the figure heading.

Sections 3.2 does not seem like results, but a continuation of the description of the study site. I do not think it belongs in the results.

Discussion

I do not think the authors should add results in the discussion. For example, new information on the dams that retain sediments. This is interesting, but it belongs to the results. Also, figure 9 does not clearly show the dams, and where sediment was trapped. Please redo. In addition, if the dams are to be considered, then the whole article should be re-structured, because initially it is supposed to focus on coastal infrastructure, not dams.

The erosion and accretion on Malva-rosa and Cabanyal cannot be seen in Figure 8. Also, this belongs to the results and not the discussion. The same applies to figure 9 and its corresponding text. They too belong to the result.

Section 4.3… the discussion around the dams and climate change is not based on the results of the study, so it does not belong here.

Conclusions

They are too long, and do not directly address the most relevant findings of the study. The concluding section seems more like another introduction.

Author Response

(The authors gave the same response as above.)

Round 2

Reviewer 3 Report

The authors have improved their article. However, the text still needs editing by a native English speaker, trhoughout the text.

Detailed comments follow.

Abstract, L8.- Do infrastructures have behavior? It would be better to say functionality, perhaps.

The goals of the study are clearly stated in the introduction.

The methods and results are largely improved.

In section 4, scientific discussion, I do not think the authors should mention “environmental economics, since this is the only time the idea is addressed. I recommend deleting this phrase. Also, what is a common beach?

I could not find the gauging stations in figure 9. Please indicate clearly where they are.

I do not see the erosion in Malva rosa (figure 10). In fact, tye beach seems quite broad here.

The conclusions are too long! This section still contains some results. They need to be condensed and focus on the take home message. I recommend the concluding section

To be 1 paragraph long. Section 4.2 seems more like results than discussion. In my opinion, the first and last paragraphs in the concluding sections would be an adequate conclusion of the article.

Line 373- change Figura for Figure

Author Response

The authors have improved their article. However, the text still needs editing by a native English speaker, trhoughout the text.

            Thank you so much. Manuscript has been submitted to MDPI English editing. Enclosed you will find service certificate.

Detailed comments follow.

Abstract, L8.- Do infrastructures have behavior? It would be better to say functionality, perhaps.

            You are right. We have changed it

The goals of the study are clearly stated in the introduction.

            Thank you so much.

The methods and results are largely improved.

            Thank you so much.

In section 4, scientific discussion, I do not think the authors should mention “environmental economics, since this is the only time the idea is addressed. I recommend deleting this phrase. Also, what is a common beach?

            Thank you so much. “common beach” was a mistake. We have deleted “environmental economics”. Finally we have written: “…clarifie  environmental effects in managing port occurring at a beaches with intensive use by the population.

I could not find the gauging stations in figure 9. Please indicate clearly where they are.

            Done. We have included situation of dam and their gauging stations in the figure

I do not see the erosion in Malva rosa (figure 10). In fact, tye beach seems quite broad here.

              You are right. Erosion on the beach of La Malvarrosa cannot be seen directly in the aerial photo, but it does exist since it does not receive sedimentary contributions from the north.

The conclusions are too long! This section still contains some results. They need to be condensed and focus on the take home message. I recommend the concluding section

            Following your comments Conclusions section has been shortened

To be 1 paragraph long. Section 4.2 seems more like results than discussion. In my opinion, the first and last paragraphs in the concluding sections would be an adequate conclusion of the article.

            Following your comments Section 4.2 has been changed

Line 373- change Figura for Figure

            Done. It was a mistake

This manuscript is a resubmission of an earlier submission. The following is a list of the peer review reports and author responses from that submission.